# Challenges Facing Leaders in Transforming Small-Scale Irrigation Farming in Usa River Ward, Arumeru District, Northern Tanzania

**Eliningaya J. Kweka** [1,2,3,*] **, Casmir F. Kitula** [1]**, Elias E. Mbuti** [1] **and David Wanani** [1,4]

1   Institute of Accountancy Arusha, Directorate of Postgraduate, Njiro Hill, Arusha P.O. Box 2798, Tanzania
2   Department of Medical Parasitology and Entomology, Catholic University of Health and Allied Sciences, Mwanza P.O. Box 1464, Tanzania
3   Pesticides Bioefficacy Unit Tanzania Plant Health and Pesticides Authority, Arusha P.O. Box 3024, Tanzania
4   Faculty of Theology and Religious Studies, University of Arusha, Usa-River, Arusha P.O. Box 7, Tanzania
*   Correspondence: pat.kweka@gmail.com

**Abstract:** The low agricultural productivity of key crops and food insecurity continue to be a problem in sub-Saharan Africa (SSA), and Tanzania in particular. The growing population and climate change further increase the food shortage. Irrigation has been strategized to reduce poverty and food insecurity, and improve the livelihoods of communities in African countries, and in particular Tanzania. Transformational leadership for small-scale irrigation is urgently needed to attain the planned agenda for irrigation schemes. This study assessed the challenges of leadership in leading and transforming small-scale irrigation schemes. The questionnaires were distributed to leaders of the agriculture sector in four strata (agriculture extension officer (25), AMCOS leaders (6), agriculture engineers (2), irrigation committee (9)) with a total of 42 leaders as participants. A total of 118 farmers were interviewed from four irrigation canals (Ngollo (32), Ngarasero I (32), Ngarasero II (33), and Abisinia (21)) in the Usa River ward. The study found that the challenges of leaders in leading the transformation of small-scale farming for success were commitment of leaders, market chain, pest control mechanisms, irrigation extension service, planning, technological transformation and adoption, mobilization of farmers and professionals, monitoring and evaluation, knowledge of irrigation, and agro-input supply. This study shows that leaders' transformation skills can play a great role in poverty reduction in small-scale irrigation in the Usa River ward. Therefore, leaders in the study area should play the role of transformational leadership effectively in managing small-scale irrigation by practicing a participatory approach to farmers problem-solving.

**Keywords:** irrigation; farmers; leaders; leadership; productivity; small-scale



## 1. Introduction

In sub-Saharan Africa (SSA), agriculture is considered the key to economic development in most countries, including Tanzania. In the history of SSA, agriculture has played a vital role in the employment of more than 80% of the population and remained the highest contributor to the gross domestic product (GDP). Agriculture in Tanzania is widely rain-fed and the irrigation practiced is merely traditional irrigation [1,2]. The development of the irrigation system in most of the areas in Tanzania is still at a lower scale than was expected since the inception of small-scale traditional irrigation in 1935 [3,4].

In Tanzania, there has been massive development of policies and programs aimed at accelerating the growth of the agriculture sector in irrigation to achieve the target impact on food security, income generation, and reliable employment for the youth population [5–8]. The government of Tanzania is scaling up the irrigation programme from small-scale to large irrigation farms to ensure food security, raw materials for industries, and foreign currency generation from exports [9,10]. The development of sustainable agricultural

irrigation growth in Tanzania has a number of hurdles facing farmers, hindering the growth of the sector [11–13].

The irrigation sector has faced a number of problems such as poor response to technology adoption, market access, leadership, persistent use of traditional schemes, and low levels of access to agro-inputs for farmers when they are needed [14–17]. In most of the areas practicing small-scale agriculture, agricultural input provisions such as the fertilizer, frequent training, pesticides, improved seeds, and small-scale irrigation have shown high yields and increases in food security in Asian and Latin American countries [17–19].

In Tanzania, agricultural sector reforms have taken a new shape with the formulation of new strategies including recruitment of agricultural extension officers countrywide, and capacitating them with the transport to reach farmers [20]. Also, there is a new agenda called the 10/30 agenda, which is detailed on transforming agriculture traditional practices into business models. The Tanzania national agriculture reform agenda and development goals have often increased agricultural productivity as a proper way to ensure national food security. However, in the past two decades, there was no progress witnessed in productivity (though rain variation was considered to be the factor) [21–23]. Despite the large area of arable land, Tanzania has not produced enough irrigation schemes comparatively to neighboring countries. Since 2011, the irrigated rice production average yield in Tanzania has been 2.0 metric tons/hectare [24], which is less than rice production in Kenya of 4 metric tons/hectare [25] and 6.7 metric tons/hectare for China [26].

For the first time since independence in 1961, the government of Tanzania has set aside a budget worth TZS 927,000,000,000 (USD 396,162,081.04) for the Ministry of Agriculture for the financial year 2022/2023. Most of the funds have been channeled into seed production, irrigation, and agro-inputs supply (fertilizer and pesticides) subsidies [27]. Despite all these efforts, small-scale irrigation farming is still not producing to the expected rates. Currently, small-scale irrigation is not achieving the expected production for food security, regardless of the availability of stable surface water.

This study investigated the leadership challenges in leading small-scale irrigation transformation in Usa River ward, Arumeru District in northern Tanzania.

## 2. Transformational Leadership and Agriculture

The transformational leadership style gained attention in the past thirty years since its inception. There are four main dimensions in transformational leadership theory that describe leaders behavior [28]. These four dimensions are (i) influence of a leader, whose charismatic behavior inspires the followers to build trust and, hence, share the mission of their leaders; (ii) inspirational motivation is the ability to formulate clear goals to share, a compelling vision that motivates the followers and promotes their expectations; (iii) the intellectual stimulation is the ability to motivate followers in questioning assumptions and proactively look for their solutions; and (iv) the individualized consideration in which leader identifies, understands, and addresses the followers developmental needs and attends to them in a timely fashion.

The permeative focus on the transformational leadership style looks more well-founded on its effects on follower's behavior and attitudes [29] and across cultures [30,31]. Due to these well-established factors, there has been a need to make clear the limits and means through which transformational leaders foster motivated followers' work outcomes, which has added attention in the leadership literature [32]. The previous findings set boundaries and mechanisms with which transformational leaders enhanced followers' performance and achievements [33–37].

To date, studies have shown great success in crystalizing the means through which the transformers can motivate and impact followers to perform beyond expectations. The previous findings by Avolio and others show that leaders pay attention to the underlying psychological processes, mechanisms, and conditions that the transformational leaders use to motivate their followers to high level of achievements and performance [32].

The previous studies conducted identified transformational leadership [38–40] in particular, and leader–follower interactions at the micro-level [41,42], as an important parameter in leadership performance, which also increase innovation. Nevertheless, the findings connecting transformational leadership and followers performance are shown to have inconsistencies [38,43]. Another theoretical approach explains that there are more mediators who link transformational leadership to innovative behavior. However, it is more important to identify the potential moderators to further provide a knowledge for better understanding of why not all followers are motivated to innovate, but also the conditions necessary for such an effect. The investigation to understand the challenges facing leaders in leading small-scale irrigation to attain production transformation is of the upmost importance, as previous findings show that transformational context has a great impact on followers' behavioral change [41,44,45]. The transformational leader guides the followers to work towards the mission, vision, and sustainability of the irrigation schemes performance [46]. The sustainability of irrigation schemes needs followers who have been well-mentored and inspired by leaders for high productivity.

## 3. Method and Material

### 3.1. Study Area Description

The study area is located at the footstep of the slopes of Mount Meru in Arumeru District, Usa River ward. The study area is comprised of four main irrigation canals, namely, the Ngarasero 1, Ngarasero 2, Ngollo, and Abisian canals. Each canal serves about 200 registered small-scale irrigation farmers. The mainly irrigated crop is paddy, but maize, beans, and horticulture are also produced. The main ethnic groups on the area include the Meru, Chagga, and Pare, with minorities such as Iraqw and Sonjo. They also practice zero grazing and poultry keeping. The data collection sites during the study are shown in Figure 1, where questionnaires were given to leaders (Meru district council and agriculture marketing cooperative society (AMCOS) office) and interviews were conducted with farmers (in the Abisinia, Ngarasero I, Ngarasero II, and Ngollo schemes).

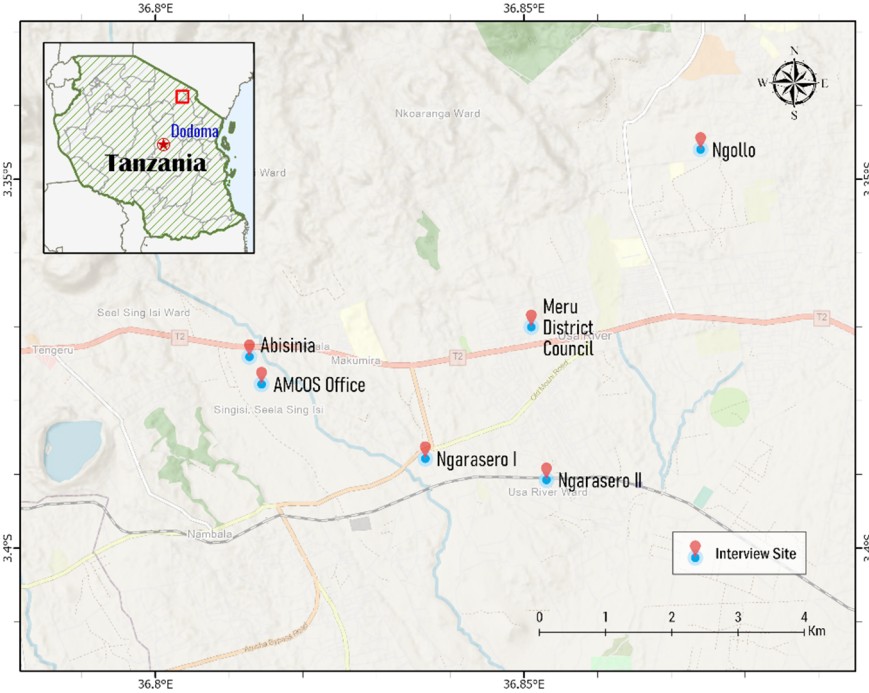

**Figure 1.** The map of the study site showing where the questionnaires were filled by leaders and interviews were conducted with farmers.

### 3.2. Study Population

The study used the population of government employee leaders who coordinated or supervised the irrigation activities in Meru District. These were agriculture extension officers, irrigation engineers, agriculture marketing cooperative society (AMCOS) leaders, and the irrigation committee members, who numbered 42 in total. The farmers involved in this study were 118 out 130 expected, who were obtained from the sample size calculation using the equation $N = n/((n + 1)(e)^2)$ (Figure 2); where N is the total number of individuals to be involved in the study, n = number of people involved in small-scale irrigation, and e = significant level (0.05).

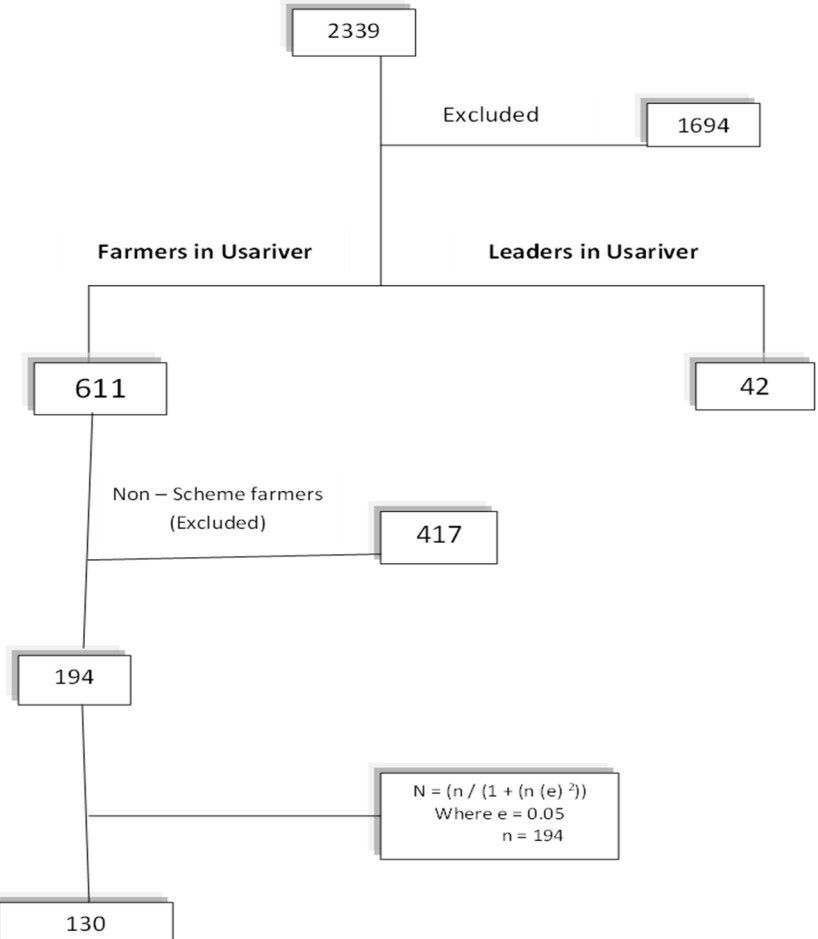

**Figure 2.** Sample size derivation flow chart (leaders, N = 42, farmers, N = 130).

### 3.3. Research Approach

The leader's groups were provided with the questionnaires to rank the main four area of concern on what are the challenges in leading the SSI in Usa River ward. These generated quantitative data. On the other hand, similar objectives with open-ended questions were used to interview the farmers who were visited either at home or on the field. These interviews generated qualitative data.

### 3.4. Data Analysis

The quantitative data were entered in an Excel sheet and transformed to SPSS version 26 (IBM Corp., Armonk, NY, USA). The data were analyzed using Kruskal–Wallis test, the non-parametric version of ANOVA as data were not normally distributed. The *p*-value was considered when the p-value was lower than 5%. For the qualitative data, the data were

pooled and presented in narrations. The Figure 2 map was prepared with ESRI ArcGIS Pro 3.0 and the map datasets from OpenStreetMap, ESRI Living Atlas, and field survey results.

*3.5. Ethical Issues*

This study was given ethical approval from the directorate of postgraduate studies at IAA (Ref. No. MBA-LG/0002/2021). The access to farmers at Usa River ward was granted by the Arumeru district executive director (Ref. No. N10/5vol VII/198). Each farmer provided written consent and was informed that participation in the study was voluntary.

## 4. Results and Discussion

*4.1. Response Rate and Demographic Data of Respondents*

A total of 172 questionnaires were dispensed. Out of those, 42 (24.4%) were given to leaders through the heads of sections in each stratum. The response rate of leaders was 100% from each stratum (Table 1). It was planned to interview 130 farmers, with an equal number of respondents of 35 farmers from each canal. The farmer's response was 90.7% (Table 1). Some farmers dropped from the study as the participation was voluntary. The demographic data of leaders show that 59.5% are males (Table 2), while for farmers, the demography shows that 64.4% are males (Table 2). In the education category, most leaders (42.9%) have diplomas, while the majority of the farmers (84.7%) have primary education (Table 2). The majority of responsive leaders (45.2%) are in the age bracket of 31–40, while 36.4% of farmers are above 50 years of age. In experience, the majority of leaders have experience of 4 to 10 years, while farmers have over 15 years of experience.

**Table 1.** Leaders and farmers response to small-scale irrigation leadership challenges.

|  | Stratum | Distributed | Returned | % Response |
|---|---|---|---|---|
| 1 | Irrigation engineer | 2 | 2 | 100 |
| 2 | Extension field officers | 25 | 25 | 100 |
| 3 | AMCOS | 6 | 6 | 100 |
| 4 | Irrigation committee | 9 | 9 | 100 |
| 5 | Farmers | 130 | 118 | 90.7 |
|  | TOTAL | 172 | 160 | 93.0 |

**Table 2.** Demographic distribution of leaders from different strata in Usa River ward (N = 42).

|  |  | Agriculture Engineer | Agriculture Extension Officer | Irrigation Committee | AMCOS |
|---|---|---|---|---|---|
| Sex | Male | 2 | 11 | 8 | 4 |
|  | Female | 0 | 14 | 1 | 2 |
|  | Total | 2 | 25 | 9 | 6 |
| Age (years) | 18–30 | 0 | 3 | 0 | 1 |
|  | 31–40 | 1 | 14 | 3 | 1 |
|  | 41–50 | 1 | 3 | 1 | 3 |
|  | >50 | 0 | 5 | 5 | 1 |
|  | Total | 2 | 25 | 9 | 6 |
| Education Status | Primary | 0 | 0 | 7 | 3 |
|  | Secondary | 0 | 0 | 2 | 1 |
|  | Certificate | 0 | 2 | 0 | 1 |
|  | Diploma | 0 | 17 | 0 | 1 |
|  | Degree | 2 | 6 | 0 | 0 |
|  | Masters | 0 | 0 | 0 | 0 |
|  | Above masters | 0 | 0 | 0 | 0 |
|  | Total | 2 | 25 | 9 | 6 |
| Work Experience (Years) | Less than 3 | 0 | 0 | 2 | 1 |
|  | 3 to 4 | 0 | 3 | 0 | 0 |
|  | 4 to 10 | 1 | 13 | 3 | 2 |
|  | 11 to 15 | 1 | 4 | 1 | 0 |
|  | More than 15 | 0 | 5 | 3 | 3 |
|  | TOTAL | 2 | 25 | 9 | 6 |

### 4.2. Commitment of Leaders

The findings on leaders commitment in this study show that the lack of leaders' commitments has a rank of median value of 3, which is under the category of undecided (Table 3). This shows that the lack of commitment in leadership discourages small-scale irrigation transformation progress. The statistical analysis shows that the response of leaders from different strata is similar, with no significant difference among them ($\chi^2 = 2.29$, df = 4, $p = 0.515$). According to 64 (54.2%) farmers interviewed (source: farmers from Ngarasero I and Ngarasero II interviewed on 12–13 July 2022), leaders have a problem of commitment to SSI transformation, they always winding with the windfall activities, and they do not put a great emphasis on small-scale irrigation challenges from farmers to enhance the transformation of SSI. Also, leaders are considered as not facilitating the access to seeds, fertilizer, and pesticides, or the repair of water canals on time. In addition, they do not negotiate for the irrigation water costs per acre, as they are too high to meet (TZS 55,000 for members and TZS 75,000 for non-members). This is similar to a previous study that assesses the technologies adoption in agriculture improvements, which found that leaders are a major component in facilitating or hindering the acceptance of technology by farmers [14,17]. The data from the document review show that leaders in small-scale irrigation do not have a strategy to subsidize agro-inputs (fertilizer, seeds, and pesticides) and make them available upon farmers' needs. The agro-inputs seem to be out of reach for many farmers when they are in high demand, due to high prices in free markets where the regulation of prices is not controlled by the government [47]. Elsewhere, it is found that farmers are claiming the costs of agro-inputs provided by micro-finances have high interests when paid, hence, making agriculture in irrigation system unsustainable, as there is no adequate financial support or soft loans access [14,17].

**Table 3.** Ranking of leaders' responses to the challenges facing small-scale irrigation.

| Factor | Median | Interpretation |
|---|---|---|
| Available market chain | 3 | Undecided |
| There is no shared vision | 3 | Undecided |
| There is no commitment from leaders | 3 | Undecided |
| Poor quality of irrigation extension service | 4 | Agree |
| Poor attitude and knowledge | 3 | Undecided |
| No technological transformation | 3 | Undecided |
| Religious and traditional barriers | 2 | Disagree |
| No monitoring and evaluation system | 3 | Undecided |
| There is poor planning | 2 | Disagree |
| Leaders are poor at fighting harmful traditions | 2 | Disagree |
| Leaders are poor are mobilizing farmers and professionals | 3 | Undecided |
| Inputs are not available on time | 4 | Agree |
| No integrated pest control mechanism | 4 | Agree |

### 4.3. Market Chain

The study demonstrates that the lack of reliable access to a market chain for small-scale irrigation crops is a profitability and productivity barrier, and affects opportunities in small-scale irrigation schemes in the Usa River ward (Table 3). The data analysis output shows that there is no significant difference in responses among leaders on market chain access for small-scale irrigation famers crops ($\chi^2 = 3.02$, df = 3, $p = 0.389$). This means that the access to profitable markets for farmers crops is limited. However, they express the thought that they have to sell the harvested crops quickly after harvest at a cheap price to pay the debts of those who lend them agro-inputs (fertilizer, seed, and pesticides), which is hampering their success. This study has similar findings to previous studies on rice trading in Tanzania and Zimbabwe, which found t several barriers impacting on price and potential customers accessibility [48–50]. The main hindrance to market access was reported previously as imported rice having lower prices than the locally produced rice [51–53]. The price difference are a decisive factor for the traders and end-users in

purchasing the foreign-produced rice from different international markets. Due to high costs on transporting locally produced rice, the imported rice seems, cheaper regardless of tariffs imposed by the government of Tanzania [53,54].

### 4.4. Pest Control Mechanism

In this study, the lack of supply of pest control (pesticides, fungicides, and herbicides) is found to have the median value of 4, which means that is under the category of agree (Table 3), showing that there is crisis in the availability of pest control pesticides. The statistical analysis in pest control mechanism shows that there is no significant difference among leaders in response to pest control mechanisms available ($\chi^2$ = 1.38, F = 3, $p$ = 0.710). This shows that, collectively, leaders agree to have pest control management in SSI farms. From this result, it is understood that the lack of supply of pesticides for small-scale irrigation is the challenge for leaders in the Usa River ward in leading small-scale irrigation productivity.

The 118 farmers who responded to interviews said that the absence of alternative solutions for pesticide sources loses them productivity, with a high loss during any pest infestation (source: farmers interviewed in Ngollo, Ngarasero I and II, and Abisinia canal on 9 July 2022). They also state that a major bottleneck for small-scale irrigation in the Usa River ward is lack of leadership on making decisions on pesticide access when they are highly needed for insect and pest control. This is similar to previous studies in Tanzania, which found that a lack of pesticides on time leads to great loss of crop harvest [17,55]. The lack of transformational leaders, according to farmers, makes it difficult for famers to effectively alleviate the challenge that leads to poverty. This is similar to studies conducted on the relationship of transformational leadership with natural performance on agribusiness [56]. This study shows that transformational leadership is a required skill among leaders to revolutionize the small-scale irrigation into a profitable agriculture system for the community.

### 4.5. Irrigation Extension Service

This study finds that the irrigation extension services have a median value rank of 4, which means that they are under the category of agree (Table 3). The statistical analysis of the response shows that there is no statistical difference among the leaders' strata ($\chi^2$ = 1.15 F = 3, $p$ = 0.765). This means that the extension services offered in the Usa River ward are not satisfactory to farmer's needs, or among leaders either. The key informant interviewed expresses that there is insufficient water-saving technology, and they have a problem of using improved seeds that are not accessible. As there are no permanent constructed canals, only the traditional canals, this causes high water loss and leads to conflicts due to water scarcity (source: farmers from Ngarasero I and Ngarasero II interviewed on 12–13 July 2022). The extension services in plant diseases are not adequate, and farmers diagnosis by themselves, which are obstacles to productivity, income, and food security. Similar studies conducted previously show that appropriate extension services to farmers decreases pests and improves food productivity and security when leaders play their role well [17,57–59]. The extension services are an invaluable service to help farmers to adapt to improved practices and technologies [14,59–62]. Technology adoption among small-scale irrigation farmers (mostly rainwater harvest) is of paramount importance for services to be delivered by extension officers among small-scale irrigation farmers, in order to ensure water availability and increase productivity for food security. The adoption of rain-water harvest technologies has influenced irrigation productivity in Kenya [63], Rwanda [64], and in Nzega, Tanzania [65], while in South Africa, there was household income increase due to the adoption of water-harvesting technologies [66,67].

The leadership attitudes in small-scale irrigation play a major role in motivating productivity when all other factors are constant. Currently, the irrigation budget in Tanzania for financial year 2022/2023 has been increased to TZS 362,000,000,000 (USD 154,828,193.00), which is 38.9% of the total Ministry of Agriculture budget [27]. The sixty-five interviewed farmers emphasize that leaders lack the ability to expand and improve the small-scale

irrigation due to the very limited resources provided by the government (source: farmers from Ngarasero I and Ngarasero II interviewed on 12–13 July 2022). The lack of transformational leadership skills among leaders handicaps their performance in leading small-scale irrigation schemes to success. In previous studies conducted elsewhere in small-scale irrigation, it is revealed that leaders with both formal and informal skills in irrigation allow their followers to perform better, hence, increased productivity and food security [68,69]. Whatever little inputs are received are careful handled and used, as both farmers and leaders are involved in the planning and implementation of the programs [68,69]. The programs conceived together by leaders and farmers based on end-user needs allow both farmers and leaders to be creative and innovative for SSI success, as previously shown in other studies [70]. The other studies findings show that the use of natural resources (water) and the traditional canal monitored well by farmers reduces water loss and facilitates high productivity [71].

The other findings of this study show that there is no shared vision between farmers and leaders in different strata, which make development programs harder to implement. There is a significant difference among leaders on how they perceive the vision-sharing between leaders and farmers in all four strata ($\chi^2$ = 11.55, F = 3, $p$ = 0.009). This shows that there is no vision-sharing among leaders and followers. When there is shared vision between leaders and followers, the working environment becomes good, while innovation and productivity increase when everybody is motivated [72,73]. A leader is referred to as the best influencer for the success and team effectiveness of farmers in the SSI [73]. This gives the best practice of transformational leading through vision-sharing and team goal commitment, which influences the transformation and mind-set of followers [74].

*4.6. Planning*

According to the findings of this study, the median value of planning for leaders in leading the small-scale irrigation in the Usa River ward is found to be 2, which ranks as disagree (Table 3). The findings among leaders of different strata have no statistical difference ($\chi^2$ = 3.65, F = 3, $p$ = 0.301). This finding indicates that leaders in Usa River small-scale irrigation and agriculture extension officers are good in planning and budgeting, but worse at implementing the plan. They express that planning without implementation is useless, therefore, they have a problem of implementing the plan due to resource scarcity. Previous studies conducted in other areas on the impact of planning on small-scale irrigation success show that proper planning of land size, previous adaptation experience, and credit access are positively associated with income generation and the stability of small-scale irrigations [75,76].

These results show that leaders working on solving farmers problems hampering small-scale irrigation will increase the ability to produce and adapt to the new positive measures, as found in other sites where leaders planning and resources where matched [58,75,76]. Leaders enhancing the availability and quality of agriculture extension education, extension services, and finance could be valuable in encouraging further farm adaptation in small-scale irrigation [75]. The small-scale irrigation leadership is all about motivating followers and implementing the plans agreed with all available resources. The farmers in small-scale irrigation in Usa River ward find that technology transformation is a challenge for them (Table 3), because the costs of technologies are not affordable. In addition, the spare parts are not available on demand when machines break down, and technical knowledge on tool repair is also a challenge for small-scale irrigation farmers (source: 65 farmers from Ngarasero I and Ngarasero II interviewed on 12–13 July 2022). The water-harvesting technologies are highly in demand as a long-term conflict-solving strategy among farmers during the dry season. Previous studies elsewhere show that the inclusion of technologies such as rain-water harvest increases the productivity in irrigation schemes during off-field rain-fed seasons [77]. It is worth calling farmers together to organize for small technologies that are affordable to start with in profitable irrigation, and perfect their usage before adopting mega-technologies.

The findings of this study reveal that the mobilization of farmers and agriculture professionals is very low (Table 3). Also, the statistical analysis shows that there is no significant difference among the leaders from all strata in response ($\chi^2$ = 6.03, F = 3, $p$ = 0.110). These findings indicate that leaders in the Usa River ward have a challenge in mobilizing the community involved in small-scale irrigation. The farmers interviewed express that there is a poor mobilization mechanism in the Usa River ward (source: the 65 farmers from Ngarasero I and Ngarasero II interviewed on 12–13 July 2022). Different studies in Asia, Africa, and Latin America show that the indigenous irrigation systems, when managed well by water users who design, build, operate, and maintain the structures of small-scale systems, participate in all project stages successes and achieve the main goals well [78]. There is a problem in integrating different tasks at the same time and mobilizing the community. Most of the time, activities are accomplished in piece-meals, not in an organized manner that helps to mobilize society morale for small-scale irrigation performance. Monitoring and evaluation of programs are vital activities; in this study, the monitoring and evaluation is found to be very low/weak in the Usa River ward.

The findings show that leaders in Usa River ward have a challenge in monitoring and evaluation, as they failed even to decide if they agree or disagree if there is no strong monitory and evaluation of the small-scale irrigation programme (Table 3). The statistical analysis shows that there is no significant difference among leaders in all strata in their response to monitoring and evaluating small-scale irrigation programs ($\chi^2$ = 1.37, F = 3, $p$ = 0.712). The sixty-five farmers interviewed also commented that leaders have no regular and strong monitoring and evaluation method. Farmers express that they have a lot of unsolved problems (source: the 65 farmers from Ngarasero I and Ngarasero II interviewed on 12–13 July 2022). Other studies show that for small-scale irrigation to take off and become profitable, monitoring and evaluation should be well executed. Based on this fact, providing farmers with information management, technical capacity, and know-how among government and non-government institutions at national, regional, zonal, and ward levels on planning, implementation, monitoring, and evaluation viewpoints related to irrigation management is a very important aspect to bring the expected result [79–82]. The studies conducted in Nigeria irrigation schemes show that the mandatory monitoring and evaluation of the schemes progress contribute to farm yield increase, and operational costs are found to be within the budgeted resources [79]. This shows that the effective monitoring and evaluation of the plans and resources allocated increases the performance of irrigation systems, and cause both leaders and farmers to be held responsible.

## 5. Conclusions and Recommendations for Small-Scale Irrigations

### 5.1. Conclusions

The findings of this study show that installation of transformational leadership skills for small-scale irrigation leaders can maximize yields, improve household income, and employment for youth. Implementation of transformational leadership skills in small-scale irrigation creates awareness, commitment, responsibilities, and accountability for both leaders and followers as they both plan, execute, and involve themselves in the decision-making process. The adoption of rain water seems to be a solution to farmers conflicts due to water shortage in the dry season.

### 5.2. Recommendations

This study lays out with a number of recommendations to all four strata of leaders (agricultural extension officer, AMCOS leaders, irrigation engineers, and the irrigation committee) involved in the Usa River ward small-scale irrigation to transform the irrigation practices and increase productivity. The study recommends the following: (i) there should be a seasonal and constant irrigation calendar to avoid conflict among farmers; (ii) quality pesticides and fertilizers should be available for the farmers before irrigation season; (iii) pest control mechanisms, plant disease, and pests are the major challenges for small-scale irrigation development. Therefore, the leaders of the Usa River small-scale irrigation

scheme should work and coordinate their challenge in collaboration with other institutions such as the Tanzania Agriculture Research Institution and the Tanzania Plant Health and Pesticides Authority; and (iv) further studies are recommended for the challenges facing leaders on leading small-scale irrigation schemes.

**Author Contributions:** E.J.K., C.F.K., E.E.M. and D.W. conceived and designed the study. E.J.K. wrote the first draft of the manuscript, review, with inputs and revisions by C.F.K., E.E.M. and D.W. All authors have read and agreed to the published version of the manuscript.

**Funding:** The manuscript was funded by a research scholarship granted to E.J.K. as part of his MBA-leadership and governance student research fund granted by Tanzania Plant Health and Pesticides Authority.

**Data Availability Statement:** All data presented in results and discussion are available upon request from corresponding author.

**Acknowledgments:** The authors highly appreciate the help given by Arumeru district agriculture leaders, AMCOS leaders, and farmers. Field assistants who interviewed farmers for questionnaires, namely, Grace Jayombo, Irene Ruta, Faustine Hosea, and Rehema Abdallah, are fondly appreciated. TPRI is acknowledged for financial support.

**Conflicts of Interest:** The authors declare no conflict of interest.

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
