# Peer review of "Challenges Facing Leaders in Transforming Small-Scale Irrigation Farming in Usa River Ward, Arumeru District, Northern Tanzania"

_2673-7655, doi:10.3390/crops2040034_

Round 1
Reviewer 1 Report
Abstract: number of interviewed farmers is mentioned in the abstract but in the study there are no reliable results related to those interviews.
Introduction: l. 74 – 78 sentences should be reformulated, the same with sentences l 106-107, 111-114. L 64-65 term Agenda is used 3 times. L 72: 2.0 metric tons/hectare, not metric/hectare.
Methods: abbrev. AMCOS is not explained. Equation l. 142 – symbols are not explained. Fig. 2 – what does the initial number represent? Statistical methods are described but in the study we do not see any output of the analysis.
Results: results of questioning of 130 farmers are only presented as a few vague statements.
In tab. 1 the number of the farmers is different to the numbers mentioned in Abstract and Methods. In tab 2. the totals in educational status are displaced. 4.3 – Market chain and following – the term produce is used in a wrong way. Sentence starting at l. 224 should be reformulated. 4.5 – the first paragraph – language revision needed. Ngarasero I and Ngarasero two – use one form of numbering.
Conclusions and recommendations: some recommendations are general, not linked to results of the study.
Language needs revision – only some examples were marked in my review.
Reviewer 2 Report
Comments:
The manuscript entitled “Challenges Facing Leaders in Transforming Small-Scale Irrigation Farming at Usa River Ward, Arumeru District, Northern Tanzania” has many grammatical issues that need to be corrected. The authors need to proofread the manuscript thoroughly. In addition, this manuscript has some ambiguous statements which do not fully depict the information provided. I suggest the following changes and improvements:
1. Authors need to revise the abstract section considering the important findings and underscore the scientific value added to your paper in your abstract.
2. What are the current research gap and the significance of this work?
3. The current structure of the introduction is not well organized and well written and the last paragraph needs to be revised, considering the main theme/objectives and findings of the study. Additionally, the novelty of this work should be stated clearly in the introduction section.
4. According to farmers (Source: farmers from Ngarasero I…………….. Need rephrasing and should be kept consistent in the text. Additionally, it will be better to write 2 sentences.
